# SS-31: A promising therapeutic agent against bleomycin-induced pulmonary fibrosis in Mice

Quankuan Gu[1,2], Yunlong Wang [1,2], Haichao Zhang[1,2], Wei Yang[1,2]*, Xianglin Meng[1,2]*, Mingyan Zhao [1,2]*

**1** Department of Critical Care Medicine, the First Affiliated Hospital of Harbin Medical University, Harbin, Heilongjiang Province, China, **2** Heilongjiang Provincial Key Laboratory of Critical Care Medicine, Harbin, Heilongjiang Province, China

* icuyangwei@sina.com (WY); mengzi98@163.com (XM); mingyan197@gmail.com (MZ)

## Abstract

### Objective

The aim of this research was to investigate if the mitochondria- targeting peptide SS-31 could serve as a protective measure against bleomycin-induced pulmonary fibrosis in mice.

### Method

Mice were split into four groups named CON group, SS-31 group, BLM group, and the BLM + SS-31 group. SS-31 (intraperitoneal injection, 5mg/Kg) was administered daily from the day prior to the experiment for the control and model groups. Mice were euthanized after 28 days of the experiment, following which blood, bronchoalveolar lavage fluid, and lung tissue were collected for analysis.

### Results

BLM caused a large decrease in body weight in mice. However, the intraperitoneal injection of SS-31 slowed down the body weight loss in the mice. It was observed through HE and Masson staining, immunohistochemistry, hydroxyproline detection, and fibrosis index measurement via Western blot that SS-31 could alleviate pulmonary fibrosis caused by BLM. Electron microscopy and ATP detection further suggested that SS-31 might help protect mitochondrial structure and function. It was also found that SS-31 could reduce reactive oxygen species and myeloperoxidase, thereby alleviating the reduction of antioxidant factor MPO and SOD, as well as diminishing the inflammatory factors TNF-α, IL-1 β, and IL-6.

### Conclusion

The mitochondria-targeting drug SS-31 exhibited potential in mitigating bleomycin-induced pulmonary fibrosis, improving mitochondrial structural and

**Data availability statement:** All relevant data are within the paper and its Supporting Information files.

**Funding:** This research was supported by Heilongjiang Province Key R&D Program(GA21C011) and the National Natural Scientific Foundation of China [82172164] and Heilongjiang Province Key R&D Program(JD22C005). The funder played a role in the design of the study, study supervision, examination of the manuscript and the publication decision.

**Competing interests:** The authors have declared that no competing interests exist.

functional damage, stabilizing the balance between oxidative and antioxidant systems, reducing inflammatory factor expression, and improving apoptosis in lung tissue.

## 1. Introduction

Fibrosis is the characteristic change and final stage outcome of numerous pulmonary diseases, including idiopathic pulmonary fibrosis (IPF), hypersensitivity pneumonitis, medical and radiation-induced lung injury, sarcoidosis, silicosis, asbestosis, cystic fibrosis (CF), and acute respiratory distress syndrome (ARDS) [1]. Pulmonary fibrosis is identified by chronic and progressive deterioration, with the average survival time post-diagnosis being 3–5 years [2]. The pathology of pulmonary fibrosis remains unclear, and the treatment options for this disease are quite limited. Besides lung transplantation, only Pirfenidone and Nintedanib have received approval for slowing the progression of pulmonary fibrosis [3].

While the pathogenesis of pulmonary fibrosis has not been thoroughly studied, it is widely accepted that injuries to the alveolar epithelial cell serve as initiators of the disease. Subsequent secondary reactions, including oxidative stress response, abnormal lung injury repair, and abnormal accumulation of the extracellular matrix, ultimately lead to pulmonary fibrosis. Oxidative stress plays a crucial role in the development of pulmonary fibrosis [4,5]. Damage to lung tissues from various causes leads to the production of a large number of reactive oxygen species (ROS) in the mitochondria, causing an imbalance in the oxidative/antioxidant system [6]. Excessive ROS enhances transforming growth factor-β (TGF-β) signaling, promoting fibrosis. This ultimately results in pulmonary fibrosis through apoptosis, triggering DNA damage, cell cycle blockade, and cell death [7]. The accumulation of dysfunctional mitochondria is considered a marker of pathological status and a key factor driving the progression of pulmonary fibrosis [8].

The SS-31 peptide is a mitochondria-targeted compound synthesized by Szeto and Schiller. It can freely cross cellular and mitochondrial membranes, selectively bind to cardiolipin through electrostatic and hydrophobic interactions, and concentrate in the mitochondrial membrane at levels over 1000 times higher than the cell matrix, thereby protecting mitochondrial structure and function [9]. Following systemic administration, SS peptide selectively binds to cardiolipin in the mitochondrial inner membrane. This interaction inhibits cardiolipin peroxidation, thereby blocking cytochrome c (Cyt c) release and mitochondrial membrane permeabilization. Consequently, it prevents mitochondrial swelling and subsequent activation of the intrinsic apoptotic cascade [10,11]. In addition to targeting mitochondria, SS-31 also cleared $H_2O_2$, HO·, and peroxidized nitrite in a dose-dependent manner to mitigate tissue-organ damage from oxidative stress [12]. SS-31 plays an important role in a variety of ischemia and reperfusion scenarios [13–15], and also mitigates damage to multiple organs caused by sepsis [16,17]. In Yang's study, SS-31 was shown to potentially improve cigarette-induced lung disease by downregulating MAPK signaling and regulating mitochondrial function, highlighting the potential positive effects

of SS-31 on the functional improvement of chronic lung diseases [18]. It has also been suggested that SS-31 can promote the metabolism of the extracellular matrix, such as matrix metalloproteinase 3 (MMP3) and matrix metalloproteinase 13 (MMP13), which are implicated in the pathogenesis of fibrotic disease [19]. In other organs, such as the kidneys and heart, SS-31 has demonstrated promising anti-fibrotic effects [20–23].

We hypothesize that SS-31 specifically targets the mitochondrial inner membrane, which is highly concentrated in alveolar epithelial cells, thereby shielding mitochondria from external stimuli. This mechanism prevents cellular oxidative stress responses and apoptotic pathways, ultimately inhibiting fibrotic progression. The aim of this research was to investigate if SS-31 could serve as a protective measure against bleomycin-induced pulmonary fibrosis in mice

## 2. Material and methods

### 2.1. Mice

C57BL/6 male mice weighing between 22–28 g were procured from Beijing Vital River Laboratory Animal Technology Co., Ltd. These mice were housed within the animal facility located at the First Affiliated Hospital of Harbin Medical University. The temperature of the rearing environment was controlled at 20–25 °C, the humidity was controlled at 40–70%, and the animals were fed ad libitum with a 12–12 h circadian rhythm. A 1-week acclimatization feeding period was provided before the start of the study.

### 2.2. Animal experiments

Mice were randomly divided into 4 groups: control (CON); control + SS-31 (SS-31); bleomycin (BLM); pulmonary fibrosis treatment group (BLM + SS-31). Each group was assigned to six mice. BLM and BLM + SS-31 groups received airway-airway bleomycin solution (instilled intra-tracheally, 3mg/Kg). The mice in the CON and SS-31 groups were infused with 0.9% sodium chloride solution. Mice in the SS-31 and BLM + SS-31 groups were injected intraperitoneally with the configured SS-31 solution (5mg/kg) every day from 1 day before model induction until day 28 after model induction. CON and BLM groups received 0.9% sodium chloride solution. All mice were weighed daily and euthanized on day 29 post-plating.

Mice were first injected intraperitoneally with lidocaine to avoid causing pain. Then pentabarbitone was injected intraperitoneally and the mice were fully anesthetized. Blood was collected after adequate anesthesia. Mice were sacrificed after blood loss. The killing methods and model establishment methods have passed the ethical standards of the Animal Ethics Committee of the above university. IACUC number: 2021115. All animal protocols were adhered to NIH requirements for activities involving the care and use of animals, and in compliance with the ARRIVE guidelines.

Mouse blood and lung tissue were collected, and about 1mm³ of lung tissue was fixed in glutaraldehyde for electron microscopy. The right lung was fixed in 4% paraformaldehyde for histopathological detection. Blood, alveolar lavage fluid and left lung of mice were stored in -80°C refrigerator for later index detection.

### 2.3. Histopathologic staining

Following infiltration, all left lung tissues from the mice were immersed in 4% paraformaldehyde for a duration of 48h. Subsequently, the tissues underwent dehydration and were embedded in paraffin. Sections with a thickness of 3 mm(Leica Biosystems, China) were then prepared. Paraffin sections were successively placed in xylene for 40min, absolute ethanol for 10min, 75% alcohol for 5min and then rinsed in tap water. Hematoxylin-eosin staining was employed to assess histopathological alterations, while Masson's trichrome staining was utilized to determine the density of accumulated collagen fibers. The severity of pulmonary fibrosis was assessed using a blinded semiquantitative Ashcroft grading system [24], The severity of pulmonary fibrosis was assessed using a blinded semiquantitative Ashcroft grading system.Quantification of collagen was processed using imageJ software(National Institutes of Health, America).

  

## 2.4. Electron microscopy

Electron microscopy using the experimental approach previously described [17]. After the mice were sacrificed, $1\,mm^3$ of lung tissue was rapidly removed and fixed in 2.5% glutaraldehyde and then subjected to dehydration, infiltration, embedding, polymerization, trimming, sectioning, and staining. After the samples were prepared, they underwent observation using a Hitachi projection electron microscope operating at 80.0 kV.

## 2.5. Immunohistochemistry

Immunohistochemistry using the experimental approach previously described [25]. Mouse lung tissues were fixed, dehydrated and embedded, antigenically repaired, endogenous peroxidase was eliminated, and the tissues were closed, subjected to incubation with primary antibody, α-SMA (1:100, CST), incubated with secondary antibody, stained with nuclei, blocked, and observed under the microscope. The α-SMA positive area was calculated for fibrosis assessment.

## 2.6. Measurement of ATP, ROS, MPO, SOD, Hydroxyproline

ATP levels in lung homogenates were measured using a commercial assay (Beyotime, China) following the manufacturer's instructions. ROS content in lung tissue homogenates was measured using commercially available ELISA kits (X-Y Biotechnology, China) in compliance with the manufacturer's guidelines. Hydroxyproline, Myeloperoxidase (MPO) and superoxide dismutase (SOD) activities in mouse lung homogenates were established in compliance with commercial norms (Nanjing, Jiancheng Bioengineering Institute, Jiangsu, China).

## 2.7. ELISA

The content of TNF-α, IL-1β, IL-6 in mouse serum was determined using the mouse-specific enzyme-linked immunosorbent assay (ELISA) kit (Jianglai Bio, China) according to the manufacturer's instructions.

## 2.8. Western blotting

Western blotting using the experimental approach previously described [25]. Protein extraction was performed as follows: Lung tissues were frozen in a -80 °C freezer were removed, and an appropriate amount was cut and dissolved in RIPA lysis buffer (Beyotime, China). Cell samples were directly withdrawn from the cell culture medium, rinsed three times with precooled PBS, and added directly into RIPA lysis buffer to collect the samples. Phenylmethanesulfonyl fluoride (Beyotime, China) was added. The tissue was thoroughly ground using a grinder, and the supernatant was extracted by centrifugation of the grinding solution at a speed of 12,000 rpm for a duration of 15min under refrigerated conditions at 4°C. Then, the protein was combined with the protein loading buffer and subjected to boiling in water at 100°C for a period ranging from 5 to 10min. SDS gel electrophoresis was performed, and the proteins were subsequently transferred to membranes and allowed to incubate for a duration of 1 to 2h. Following this, the membrane was subjected to blocking with 5% skim milk powder for an additional 1 to 2h prior to incubation with the designated antibodies: ZO-1 (1:1000 dilution, Santa Cruz), E-cadherin (1:1000 dilution, CST), COL1A1 (1:1000 dilution, Abcam), VIM (1:1000 dilution, Abcam), a-SMA (1:1000 dilution, CST), TNF-α (1:1000 dilution, Abcam), IL-1β (1:1000 dilution, Abcam), and IL-6 (1:1000 dilution, Abcam). caspase-3 (1:1000 dilution, CST), caspase-9 (1:1000 dilution, CST), BAX (1:1000 dilution, CST), BCL (1:1000 dilution, CST), GAPDH (1:10000 dilution, abcam), shaken overnight in a refrigerator at 4°C, the primary antibody was retrieved, and the membrane was subsequently washed before undergoing incubation with the fluorescent secondary antibody. Once the secondary antibody was retrieved, the membrane underwent additional washing steps before being subjected to development. Finally, we quantified the results using imageJ software(National Institutes of Health, America).

 

## 2.9. RT−PCR

Whole RNA was taken from animal lung tissue and cell samples using an RNA extraction kit (TRAN) according to the instructions. After measuring the RNA concentration, The RNA underwent reverse transcription into cDNA utilizing a reverse transcription kit. Then, 2 μl of diluted cDNA, 1 μl of each upstream and downstream primer, 10 μl of SYBR (Roche, Switzerland), and 8 μl of enzyme-free water were added to each reaction well. All reagents were mixed at the bottom of the reaction wells by centrifugation. PCR amplifications were conducted utilizing a PCR instrument manufactured by Applied Biosystems(PCR reaction: 95°C for 5 seconds, 60°C for 20 seconds. A total of 40 cycles). The primer sequences can be found in table 1.

## 2.10. Statistical analysis

All statistical analyses were conducted using R software version 4.2.2 (R Foundation for Statistical Computing, Vienna, Austria). The data were expressed as mean values accompanied by their corresponding standard deviations. Group mean comparisons were conducted using one-way ANOVA followed by the least significant difference test for multiple comparisons. A significance threshold of $P < 0.05$ was applied to determine statistical significance.

## 3. Results

### 3.1. SS-31 Mitigates Weight Loss and Inflammatory Response Induced by BLM in Mice

Daily body weight changes in the mice are depicted in Fig 1. The CON and SS-31 groups gained 8.36% and 5.88% body weight on day 28 as compared to day 1, respectively, whereas the BLM+SS-31 group also saw an increase of 2.51%. Conversely, the BLM group lost 2.67% of body weight from the first day (Table 2). This highlights that lung fibrosis induced by BLM leads to weight loss, but SS-31 mitigates this loss (Fig 1A). Mice in all groups exhibited significant weight loss on the first day post-model creation. As inhalation of BLM or normal saline can cause some lung tissue damage and affect

**Table 1. Primers used for RT-PCR.**

| Primer name | Sequence (5' -> 3') |
|---|---|
| *Tnf-α* | **Forward:** CTGAACTTCGGGGTGATCGG |
| | **Reverse:** GGCTTGTCACTCGAATTTTGAGA |
| *Il-1β* | **Forward:** GAAATGCCACCTTTTGACAGTG |
| | **Reverse:** TGGATGCTCTCATCAGGACAG |
| *Il-6* | **Forward:** CTGCAAGAGACTTCCATCCAG |
| | **Reverse:** AGTGGTATAGACAGGTCTGTTGG |
| *Zo-1* | **Forward:** GCTTTAGCGAACAGAAGGAGC |
| | **Reverse:** TTCATTTTTCCGAGACTTCACCA |
| E-Cadherin | **Forward:** CAGTTCCGAGGTCTACACCTT |
| | **Reverse:** TGAATCGGGAGTCTTCCGAAAA |
| *Vim* | **Forward:** TCCACACGCACCTACAGTCT |
| | **Reverse:** CCGAGGACCGGGTCACATA |
| *Col1a1* | **Forward:** TAAGGGTCCCCAATGGTGAGA |
| | **Reverse:** GGGTCCCTCGACTCCTACAT |
| *α-Sma* | **Forward:** CCCAGACATCAGGGAGTAATGG |
| | **Reverse:** TCTATCGGATACTTCAGCGTCA |
| *Gapdh* | **Forward:** AATGGATTTGGACGCATTGGT |
| | **Reverse:** TTTGCACTGGTACGTGTTGAT |

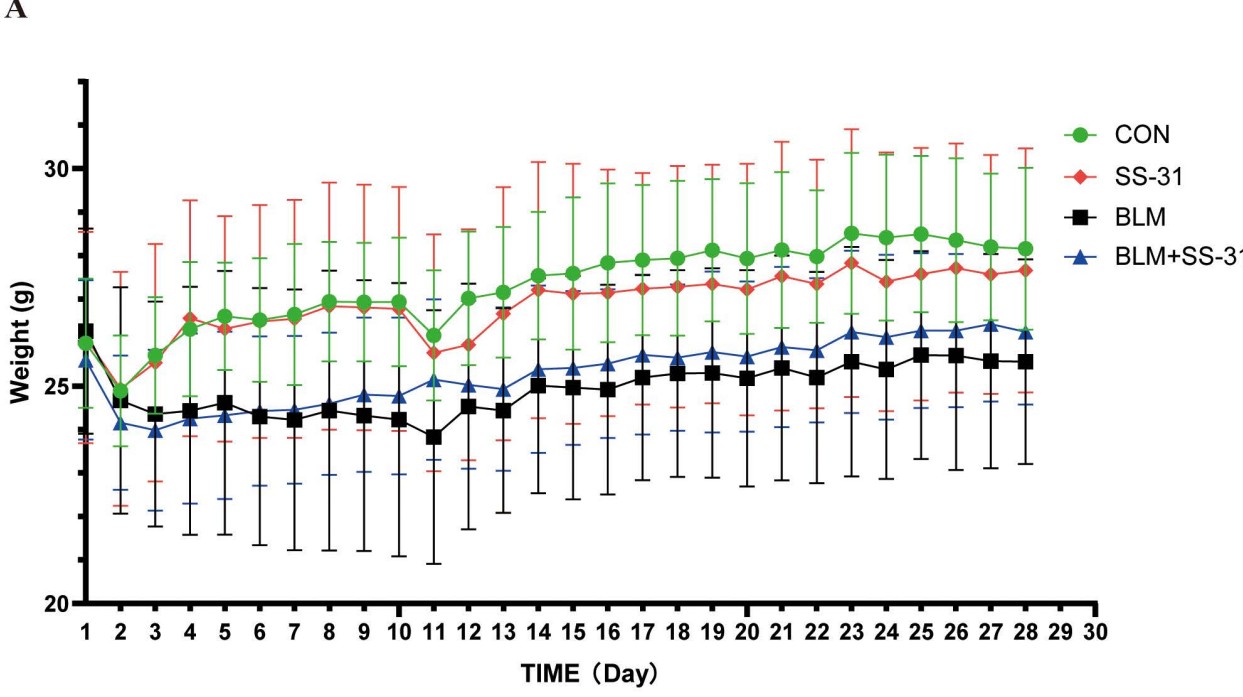

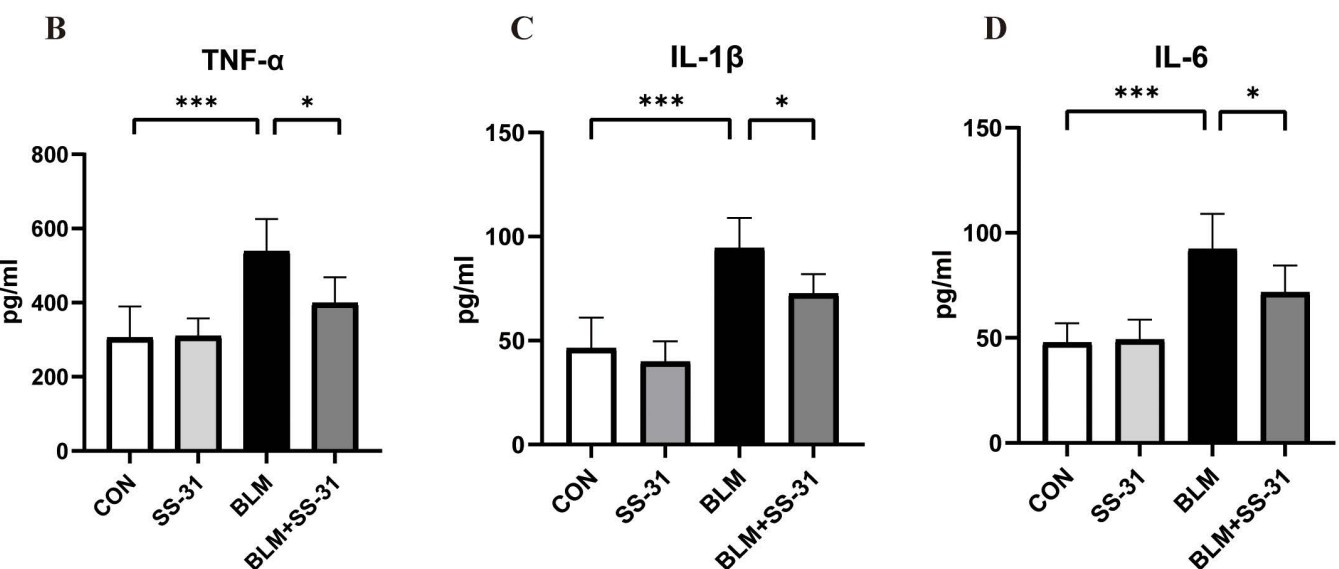

**Fig 1. Effect of SS-31 on mouse body weight and inflammatory factors in serum. (A)** Body weight trend of each group after BLM modeling(n = 6); **(B)** Content of TNF-α in serum(n = 6); **(C)** Content of IL-β in serum(n = 6); **(D)** Content of IL-6 in serum(n = 6). CON: control, BLM: bleomycin. *P < 0.05; **P < 0.01; ***P < 0.001; ****P < 0.0001.

**Table 2. Effect of SS-31 on bleomycin-induced body weight loss in mice.**

| Groups | Initial body weight (g) | Final body weight (g) | Body weight gain(%) |
|---|---|---|---|
| CON | 25.98 ± 1.48 | 28.15 ± 1.86 | 8.36 |
| SS-31 | 26.12 ± 2.43 | 27.66 ± 2.81 | 5.88 |
| BLM | 26.26 ± 2.36 | 25.56 ± 2.35 | -2.67*** |
| BLM + SS-31 | 25.60 ± 1.83 | 26.24 ± 1.67 | 2.51### |

***: CON vs BLM, $P < 0.001$; ###: BLM vs BLM + SS-31, $P < 0.001$.

the mice's appetite, this is unsurprising. However, mice in the saline group quickly regained their appetite and resumed a normal diet the next day, while those in the BLM group showed slower recovery.

We analyzed the inflammatory factors in the serum of each group of mice. Post-BLM airway inhalation, TNF-α in serum increased by 233 pg/mL, and IL-1 β and IL-6 both increased by 45 pg/mL, compared to the CON group. However, after intraperitoneal injection of SS-31, TNF-α decreased by 138 pg/mL, IL-1 β by 22 pg/mL, and IL-6 by 21 pg/mL relative to the BLM group. These results were statistically significant, suggesting that SS-31 can improve the systemic inflammatory changes in mice due to BLM (Fig 1B-D).

### 3.2. SS-31 attenuates lung tissue fibrosis induced by BLM in mice

Mice received BLM infusions for 28 days. The lung structure remained mostly intact with reduced interstitial hyperplasia and lessened inflammatory exudation post-intraperitoneal injection of SS-31 (Fig 2A). Fig 2B shows Masson staining. Collagen fibers in lung tissue, stained blue, markedly increased in the BLM group, but decreased after SS-31 injection. α-SMA, a common marker of fibrosis, expression increased in the BLM group's lungs, but decreased with SS-31 (Fig 2C). According to the Ashcroft score, the score for the BLM group was 4.2 points higher than the CON group, and the BLM + SS-31 group's score was 2.6 points lower than the BLM group (Fig 2D). The collagen area in the BLM group increased by 31% compared to the CON group, but decreased to 18% after receiving SS-31, which was 15% less than the BLM group (Fig 2E). The α-SMA positive area increased by about 25% in the BLM group compared to the CON group, but decreased by about 10% in the BLM + SS-31 group (Fig 2F). All these results were statistically significant, indicating that SS-31 can alleviate the destruction of lung tissue structure induced by BLM.

### 3.3. SS-31 Alleviates BLM-Induced Inflammation in Mice

Inflammatory factors TNF-α, IL-1 β, and IL-6 in lung tissue homogenates of each group were examined by Western blot. BLM significantly increased inflammation in the lung tissues of mice compared to the control group, but this increase was significantly reduced after SS-31 injection (Fig 3A). RT-PCR experiments revealed that SS-31 could reduce the increase of lung tissue inflammatory factors caused by BLM at the gene level (Fig 3B-D). This suggests that SS-31 can ameliorate lung tissue inflammation caused by BLM in mice.

### 3.4. SS-31 Protects against Mitochondrial and Oxidative Stress Damage Induced by BLM in Mouse Lung

Tissue Upon electron microscopic examination of mouse lung tissue, the mitochondrial size and structure in the alveolar epithelial cells of the CON group were relatively normal, with clear visibility of internal mitochondrial ridges. In contrast, the BLM group showed a marked reduction in the number of mitochondria in the alveolar epithelial cells, featuring mitochondrial swelling, disordered mitochondrial cristae arrangement, and membrane destruction. After SS-31 treatment, although the mitochondria were still more swollen than the normal group, they maintained the basic structure with orderly mitochondrial cristae and intact mitochondrial membranes (Fig 4A).

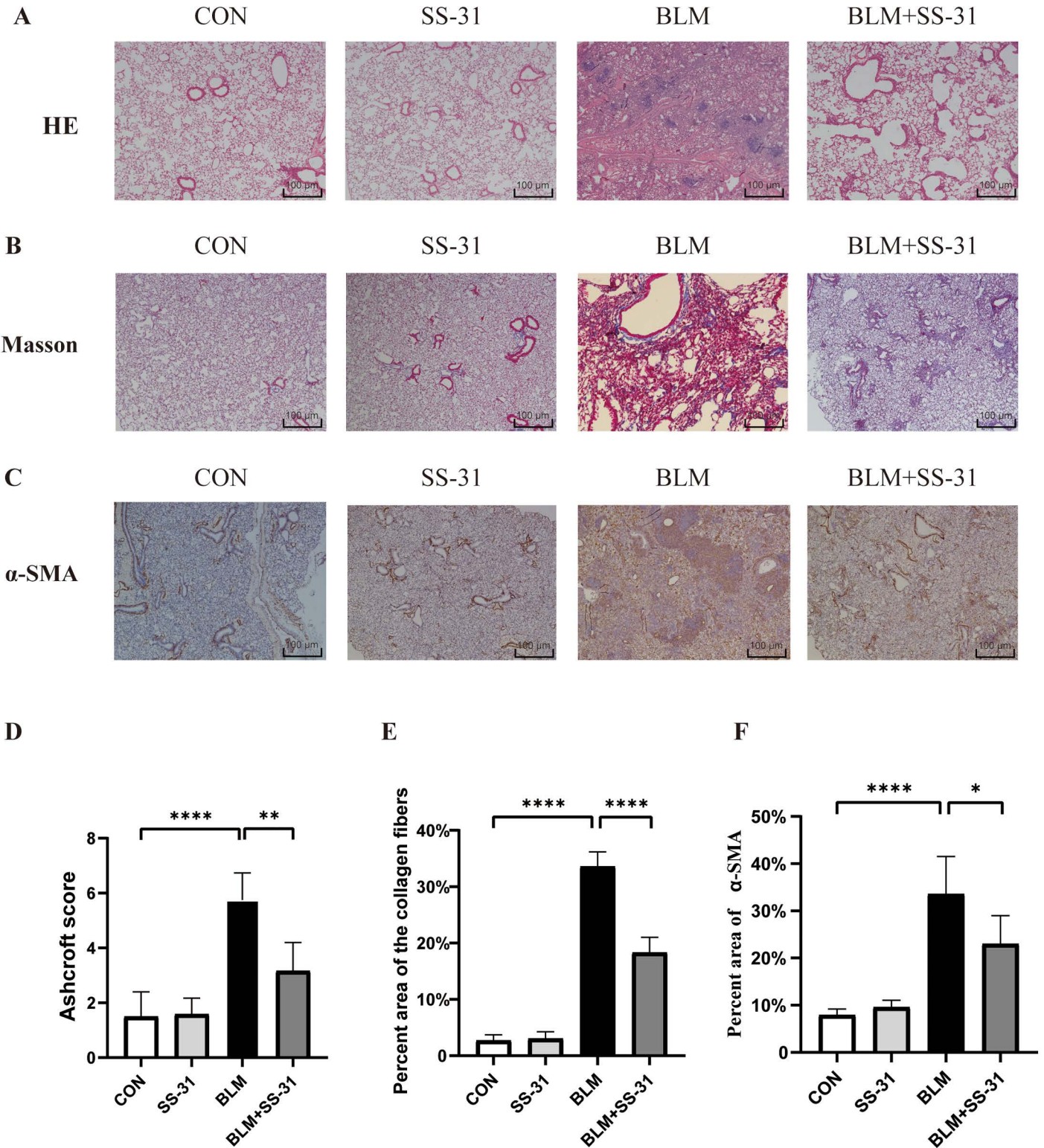

**Fig 2. Pathological staining and the pathological score. (A)**Hematoxylin-eosin staining(n = 6); **(B)** Masson staining(n = 6); **(C)** Immunohistochemistry(n = 6); **(D)** Ashcroft score of mouse lung tissue(n = 6); E: Positive area for fibrosis(n = 6); F: Positive area for α-SMA(n = 6).*$P$ < 0.05; **$P$ < 0.01; ***$P$ < 0.001; ****$P$ < 0.0001.

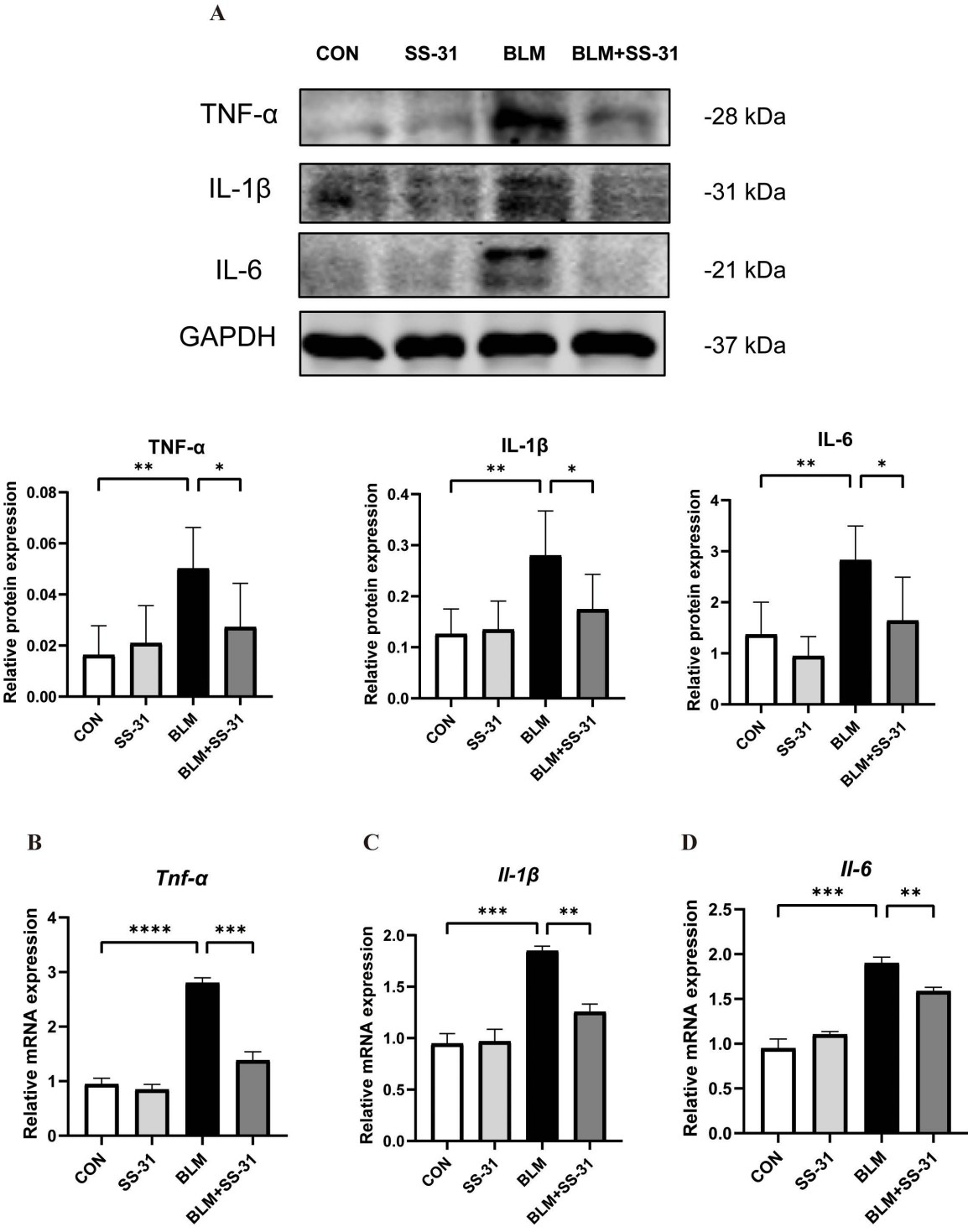

**Fig 3. Effect of SS-31 on mice lung tissue inflammation induced by BLM.** (A)Western blot detection of TNF-α, IL-1β and IL-6 changes in lung tissues and grayscale value analysis(n = 6); **(B)** Gene expression of *Tnf-a* in lung tissues measured by RT-PCR(n = 6); **(C)** Gene expression of *Il-1β* in lung tissues measured by RT-PCR(n = 6); **(D)** Gene expression of *Il-6* in lung tissues measured by RT-PCR(n = 6). *P < 0.05; **P < 0.01; ***P < 0.001; ****P < 0.0001.

**A**

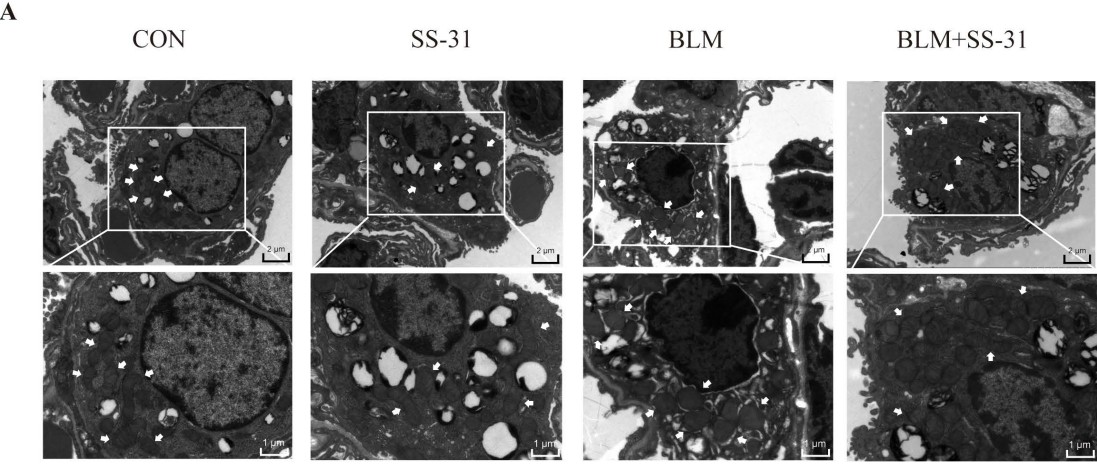

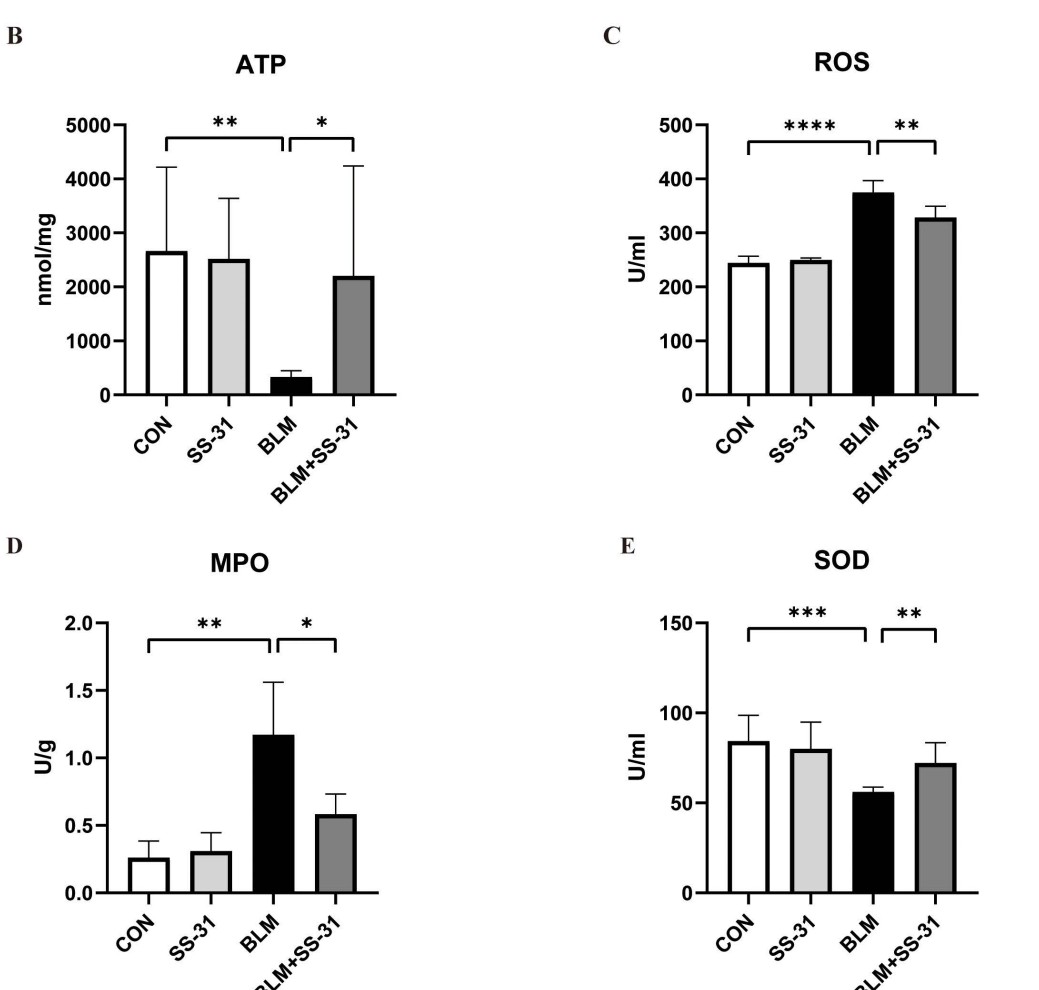

**Fig 4. Mitochondrial structural changes and oxidative stress. (A)** Transmission electron microscopy to observe the structural changes in type II alveolar epithelial cells **(B)** Changes in ATP indices in the lung tissues of mice in each group by kit(n=6); **(C)** ELISA to detect the changes in ROS in the lung tissues(n=6); **(D)** Changes in myeloperoxidase (MPO) content in the lung tissues of mice in each group by kit(n=6); **(E)** Changes in superoxide dismutase (SOD) content in the lung tissues of mice in each group by kit(n=6). *$P<0.05$; **$P<0.01$; ***$P<0.001$; ****$P<0.0001$.

Mitochondria are primary sites for ATP synthesis. The levels of ATP directly reflect the state of mitochondrial function. Upon measurement, the average ATP content of the lung tissue in the BLM group was 333.3 μmol/mg, about 2300 μmol/mg less than the CON group. After intraperitoneal injection of SS-31, ATP content recovered to an average of 2201 μmol/mg, approximately 1900 μmol/mg higher than the BLM group ($P < 0.05$). This suggests that SS-31 could improve the impairment of mitochondrial function caused by BLM (Fig 4B).

By measuring the oxidative stress index in lung tissue homogenate, the ROS content of the BLM group increased by 130U/mL compared with the CON group, while the BLM + SS-31 group saw a decrease of about 50U/mL compared with the BLM group (Fig 4C). BLM increased MPO content in lung tissue by 0.9U/g compared with the CON group, while after SS-31, MPO content decreased about 0.6U/g compared with the BLM group (Fig 4D). BLM also reduced the antioxidant stress factor SOD, but after SS-31, SOD increased to 64.830U/mL, an increase of 9U/mL compared with the BLM group (Fig 4E). These results indicate that SS-31 can improve the imbalance of the oxidative/antioxidant system caused by BLM.

### 3.5. SS-31 Reduces Pulmonary Fibrosis Markers in BLM

By using Western blot to detect COL1A1, VIM, and α-SMA in lung tissue homogenates from each group of mice, we found that BLM significantly increased the fibrosis indexes of mouse lung tissue compared to the control group. After intraperitoneal injection of SS-31, these indexes significantly decreased compared to the BLM group. ZO-1 and E-cadherin, indicators of epithelial cells, significantly decreased after BLM stimulation, while SS-31 mitigated the epithelial cell damage caused by BLM (Fig 5A). The same results were obtained from changes in these indicators at RNA levels by RT-PCR (Fig 5B-F). We also tested the content of hydroxyproline in lung tissue and found that after bleomycin, the content of hydroxyproline in lung tissue increased by 495.5 μg/g compared with the CON group, while after SS-31 treatment it decreased by 221.2 μg/g compared to the BLM group. The experiment demonstrates that SS-31 could improve pulmonary fibrosis in mice induced by BLM.

### 3.6. SS-31 attenuates lung tissue apoptosis in mice due to BLM

Following the detection of apoptosis in lung homogenates from each group via Western blot, pro-apoptotic proteins caspase-3, caspase-9, and BAX increased, while the anti-apoptotic protein BCL 2 decreased. Injection of SS-31 reduced the upregulation of pro-apoptotic proteins caused by BLM and protected anti-apoptotic proteins (Fig 6A). This suggests that SS-31 could alleviate the apoptosis caused by BLM.

## 4. Discussion

This study used bleomycin tracheal infusion to induce pulmonary fibrosis, a commonly used method for inducing this condition [26]. Pulmonary fibrosis was confirmed through histopathology, immunohistochemistry, and the detection of related biomarkers. In previous studies, much attention has been focused on the treatment of SS-31 for acute organ injury, while little attention has been paid to the treatment of fibrosis, especially pulmonary fibrosis. This experimental group used bleomycin tracheal infusion to induce pulmonary fibrosis. The advantages of BLM molding include easy induction and relatively short molding time, which is the most commonly used molding method to induce pulmonary fibrosis [26]. Through this model, we demonstrated that SS-31 can alleviate lung fibrosis in mice, improve the disruption of mitochondrial structure along with mitochondrial function caused by BLM, and adjust the imbalance of the oxidative/antioxidant system.

Oxidative stress, characterized by an imbalance between reactive oxygen species and cellular antioxidant capacity, results in the disruption of redox signaling and/or molecular damage [27–29]. Simona Inghilleri et al. found that bleomycin significantly increased the number of phagocytes and alveolar type epithelial cells, leading to an increased production of ROS compared to controls [30]. Numerous studies have highlighted the role of oxidative stress in the evolution of

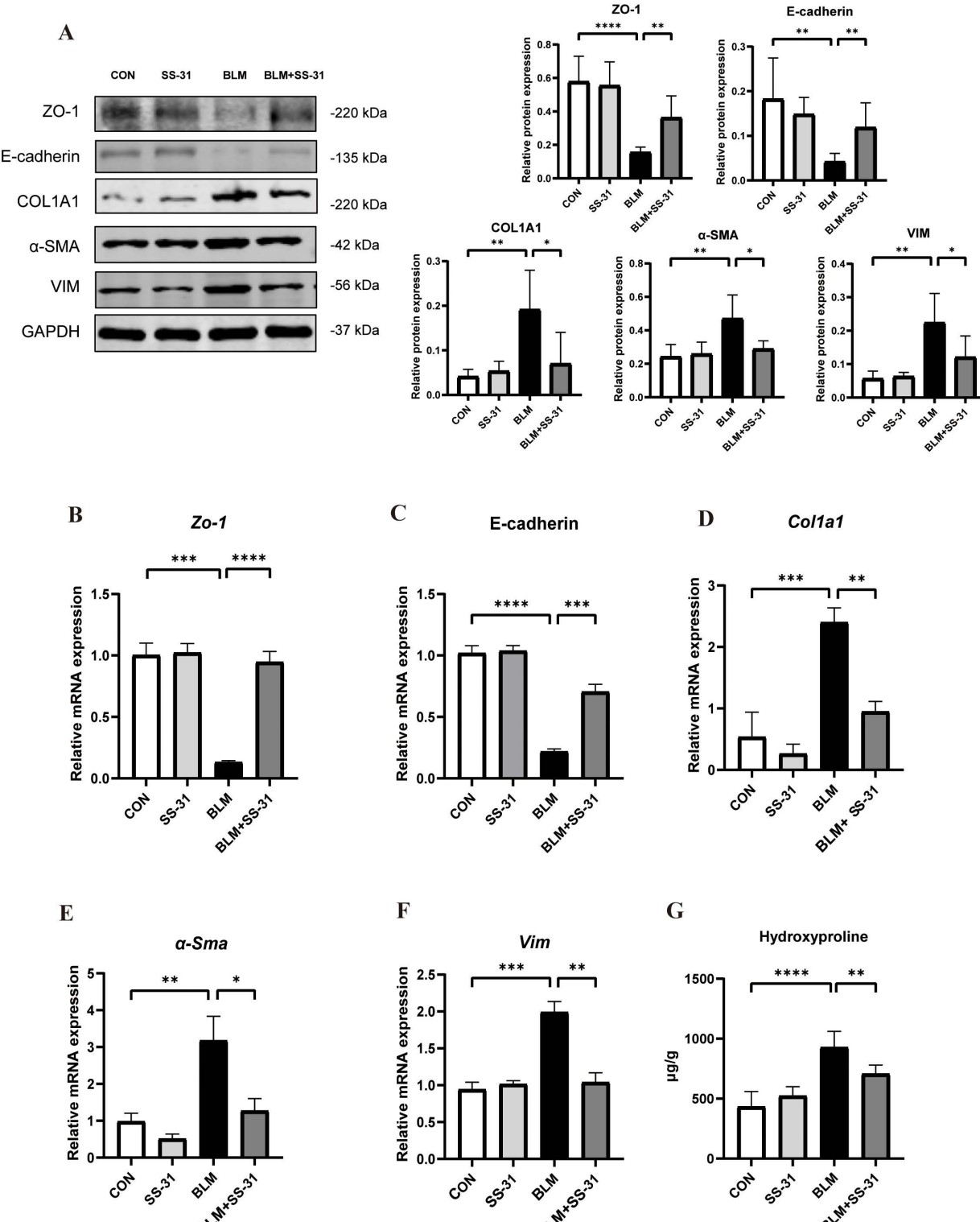

**Fig 5. Effect of SS-31 on mice lung tissue inflammation induced by BLM.** (A)Western blot detection of ZO-1, E-cadherin, COL1A1, a-SMA and VIM changes in lung tissues and grayscale value analysis(n = 6); (B~F) Gene expression of zo-1, *E-cadherin, Col1a1, α-Sma* and *Vim* in lung tissues measured by RT-PCR(n = 6); **(G)**Expression of hydroxyproline in the mouse lung tissue.*$P < 0.05$; **$P < 0.01$;***$P < 0.001$; ****$P < 0.0001$.

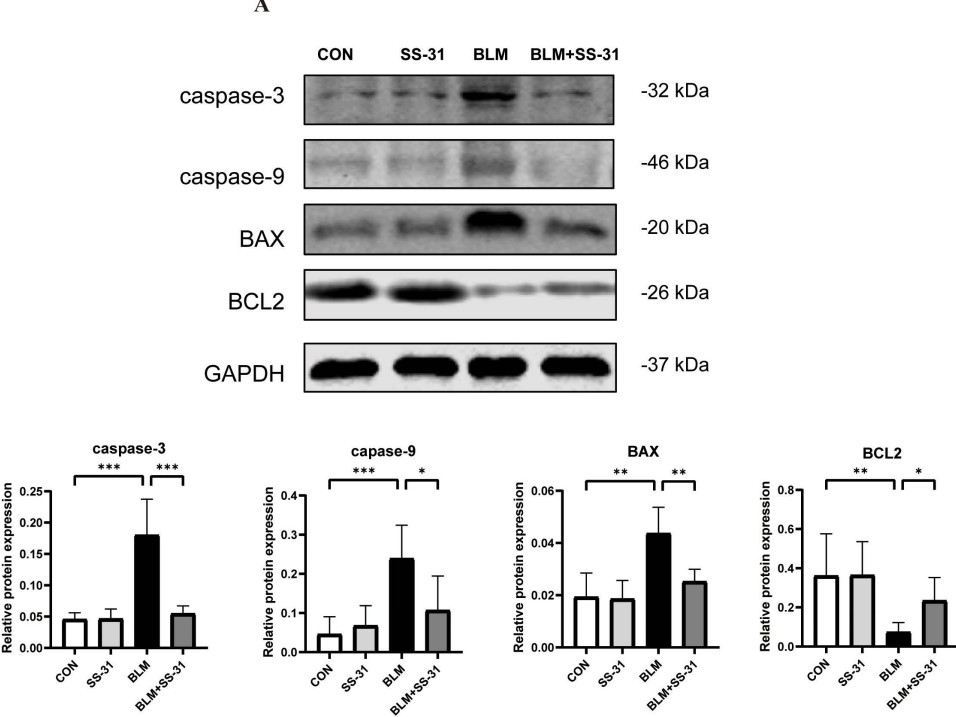

**Fig 6. Effect of SS-31 on apoptosis in lung tissues.** (A) Western blot detection of caspase-3, caspase-9, BAX and BCL2 changes in lung tissues and grayscale value analysis(n = 6); *P < 0.05; **P < 0.01; ***P < 0.001; ****P < 0.0001.

pulmonary fibrosis. In the study by Hemnes, PDE 5 inhibitors lessened bleomycin-induced pulmonary fibrosis and pulmonary hypertension by inhibiting ROS generation [31]. Talat Kilic's studies found that molsidomine demonstrated free radical clearance, antioxidant, anti-inflammatory, and antifibrotic properties [32]. Tamagawa also demonstrated the inhibitory effect of Lecithinized Superoxide Dismutase on bleomycin-induced pulmonary fibrosis, suggesting that the oxidative/antioxidant imbalance may play a crucial role in pulmonary fibrosis development [33]. The activation of the oxidative pathway in bleomycin-induced lung injury is induced not only due to the excess production of ROS, but also because of the abundance of inflammatory cells. It was found that SS-31 can reduce not only reactive oxygen species but also inflammatory factors in lung tissue, with its antioxidant and anti-inflammatory effects significantly contributing to the reduction of pulmonary fibrosis.

Mitochondrial dysfunction plays a pivotal role in the pathogenesis of pulmonary fibrosis. It has been shown that several regulatory mechanisms controlling mitochondrial function are dysregulated in IPF patient lung epithelial cells, fibroblasts, and macrophages, with mitochondria losing their normal cristae structure, becoming larger, producing less ATP, and increasing ROS [34,35]. Increased production of mitochondrial ROS, mitochondrial DNA damage, and abnormal mitochondrial quality control, are all associated with fibrotic processes, including epithelial cell apoptosis and senescence, fibroblast senescence, and immune cell dysfunction [36]. In this experiment, in response to BLM induction, the mitochondria in type II alveolar epithelial cells swelled and the mitochondrial ridges were destroyed.

This study has several limitations. Firstly, we only described the improvement of pulmonary fibrosis via some objective indicators, and we did not measure the lung function in mice. It is one-sided to suggest that mouse lung fibrosis has improved merely from weight change and other objective indicators. Secondly, this study primarily focuses on the preventive effect of SS-31 on bleomycin-induced pulmonary fibrosis, while the potential reversal effect of SS-31 on already

formed pulmonary fibrosis has not been investigated. In Yang's study, SS-31 acted by inhibiting the MAPK pathway [18]. Another study found that SS-31 alleviates LPS-induced acute lung injury by inhibiting inflammatory responses through the S100A8/NLRP3/GSDMD signaling pathway [37]. We will investigate the study of the mechanism of SS-31 in the future.

## 5. Conclusion

The mitochondrial targeting drug SS-31 can ameliorate bleomycin- induced pulmonary fibrosis in mice, improve the destruction of mitochondrial structure and function, stabilize the imbalance between oxidative and antioxidant systems, reduce the expression of inflammatory factors, and improve the apoptosis of lung tissue. Our preliminary findings suggest that the therapeutic mechanism of SS-31 in attenuating pulmonary fibrosis appears to involve downregulation of the MAPK signaling pathway, though requiring further mechanistic validation through in vivo phosphorylation profiling.

## Author contributions

**Conceptualization:** Quankuan Gu, Xianglin Meng.

**Data curation:** Quankuan Gu, Yunlong Wang, Haichao Zhang.

**Formal analysis:** Quankuan Gu, Yunlong Wang, Haichao Zhang.

**Funding acquisition:** Mingyan Zhao.

**Investigation:** Quankuan Gu.

**Methodology:** Quankuan Gu.

**Software:** Quankuan Gu, Yunlong Wang, Haichao Zhang.

**Supervision:** Wei Yang, Xianglin Meng, Mingyan Zhao.

**Validation:** Wei Yang, Xianglin Meng, Mingyan Zhao.

**Writing – original draft:** Quankuan Gu.

**Writing – review & editing:** Wei Yang, Xianglin Meng, Mingyan Zhao.

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
