## [Decision Letter · Decision Letter 0]

16 Jan 2025

PONE-D-24-54571SS-31 protects against bleomycin-induced lung injury and fibrosisPLOS ONE

Dear Dr. Zhao,

Thank you for submitting your manuscript to PLOS ONE. After careful consideration, we feel that it has merit but does not fully meet PLOS ONE’s publication criteria as it currently stands. Therefore, we invite you to submit a revised version of the manuscript that addresses the points raised during the review process.

We look forward to receiving your revised manuscript.

Kind regards,

Sairah Hafeez Kamran, PhD

Academic Editor

PLOS ONE

Journal Requirements:

“This research was supported by Heilongjiang Province Key R&D Program(GA21C011) and the National Natural Scientific Foundation of China [82172164] and Heilongjiang Province Key R&D Program(JD22C005).”

6. PLOS requires an ORCID iD for the corresponding author in Editorial Manager on papers submitted after December 6th, 2016. Please ensure that you have an ORCID iD and that it is validated in Editorial Manager. To do this, go to ‘Update my Information’ (in the upper left-hand corner of the main menu), and click on the Fetch/Validate link next to the ORCID field. This will take you to the ORCID site and allow you to create a new iD or authenticate a pre-existing iD in Editorial Manager.

7.  Thank you for stating the following in the Acknowledgments Section of your manuscript:

“This research was supported by Heilongjiang Province Key R&D Program(GA21C011) and the National Natural Scientific Foundation of China [82172164] and Heilongjiang Province Key R&D Program(JD22C005).”

“This research was supported by Heilongjiang Province Key R&D Program(GA21C011) and the National Natural Scientific Foundation of China [82172164] and Heilongjiang Province Key R&D Program(JD22C005).”

8. PLOS ONE now requires that authors provide the original uncropped and unadjusted images underlying all blot or gel results reported in a submission’s figures or Supporting Information files. This policy and the journal’s other requirements for blot/gel reporting and figure preparation are described in detail at https://journals.plos.org/plosone/s/figures#loc-blot-and-gel-reporting-requirements and https://journals.plos.org/plosone/s/figures#loc-preparing-figures-from-image-files. When you submit your revised manuscript, please ensure that your figures adhere fully to these guidelines and provide the original underlying images for all blot or gel data reported in your submission. See the following link for instructions on providing the original image data: https://journals.plos.org/plosone/s/figures#loc-original-images-for-blots-and-gels.   

**Additional Editor Comments:**

Dear Authors

Although this study is intriguing, a few issues need to be fixed before it can be accepted. Please include the study's goal at the conclusion of the introduction. Kindly add the euthanasia technique to the list of procedures.

Reviewers' comments:

Reviewer's Responses to Questions

**Comments to the Author**

1. Is the manuscript technically sound, and do the data support the conclusions?

Reviewer #1: Yes

Reviewer #2: Yes

2. Has the statistical analysis been performed appropriately and rigorously? 

Reviewer #1: Yes

Reviewer #2: Yes

3. Have the authors made all data underlying the findings in their manuscript fully available?

Reviewer #1: Yes

Reviewer #2: Yes

4. Is the manuscript presented in an intelligible fashion and written in standard English?

Reviewer #1: Yes

Reviewer #2: Yes

5. Review Comments to the Author

Reviewer #1: Review Report:

The manuscript titled "SS-31 protects against bleomycin-induced lung injury and fibrosis" suggests that the research focuses on the therapeutic potential of SS-31 in preventing or alleviating lung damage and scarring caused by bleomycin in experimental models.

Note: I reviewed the file named (PONE-D-24-54571) downloaded from journal online portal and all my comments are according to the line numberings of the pdf file “PONE-D-24-542471”

Detailed Comments/Recommendations:

Title:

 The title of the study, "SS-31 protects against bleomycin-induced lung injury and fibrosis" effectively communicates the focus of the study, specifically the therapeutic role of SS-31 in addressing bleomycin-induced lung injury and fibrosis. appears to be apt and engaging. It reflects the experimental context and potential outcomes, making it relevant to researchers interested in mitochondrial-targeted therapies or pulmonary fibrosis models.

 The title is suitable for the scope of the study, but slight refinements, such as specifying the experimental model or incorporating mechanistic details, would enhance its clarity, relevance, and appeal to the target audience.

 Suggested Title: SS-31: A Promising Therapeutic Agent Against Bleomycin-Induced Pulmonary Fibrosis in Mice"

Abstract:

 The abstract clearly states the purpose of the research, focusing on the potential protective effects of SS-31 against bleomycin-induced pulmonary fibrosis in mice.

 Lines 57-59: SS-31 was administered daily but does not specify the dosage or route of administration. Including this detail would strengthen the methodological clarity.

 Lines 62-63: The study demonstrated that SS-31 could potentially mitigate the reduction in mice" is incomplete, leaving the reader unclear about what specific reduction is being referred to.

 Lines 62-70: While methods such as HE staining, Masson staining, and Western blot are mentioned, the abstract lacks numerical or statistical details to quantify the findings. Providing key metrics (e.g., percentage reduction in fibrosis or ROS levels) would enhance credibility.

 Lines 71-75: While the abstract mentions outcomes like reduced ROS and inflammatory factors, it does not delve into the specific mechanisms through which SS-31 exerts these effects. A brief mention of the proposed pathways would be helpful.

 Address any typographical errors and ensure proper scientific notation.

Key Words: Three out of five key words are already mentioned in title. It will be more attractive if the writer mentions some other relevant key words.

Introduction:

 Combine the 1st and 2nd paragraphs to provide a concise yet comprehensive summary of pulmonary fibrosis and its challenges.

 In 2nd Paragraph: The section on pulmonary fibrosis repeats some information (e.g., limited treatments and poor prognosis). Streamlining these details can make the introduction more concise.

 In 2nd paragraph: While the role of oxidative stress and ROS in pulmonary fibrosis is mentioned, the discussion could benefit from a brief elaboration on how SS-31 specifically counteracts these processes.

 Lines 107 to onward: Include a specific mention of how SS-31 reduces ROS or preserves mitochondrial integrity, tying it directly to pulmonary fibrosis pathogenesis.

 Lines 122 to onward: Clarify the connection between pulmonary hypertension and fibrosis, or reduce the emphasis on PH if it is not a major focus of the research.

 Some references (e.g., oxidative stress role, SS-31’s effects) are summarized without detailed explanations or direct linkage to the study’s aims. Consider elaborating on key studies to provide stronger support for SS-31's potential.

Materials and Methods:

 Lines 139-141: “The killing methods and model establishment methods have passed the ethical standards...”: Rephrase to improve readability and remove redundancy.

 Lines 151-153: Terms like “molding” (e.g., “from 1 day before molding until day 28 after molding”) are unusual and may confuse readers. Use “model induction” or “bleomycin treatment” instead.

 Lines 162, Histopathology: The paraffin embedding and sectioning steps should clarify equipment or specific dehydration/embedding protocols.

 Lines 192-193: ATP content in mouse lung homogenate was determined according to commercial protocols” can be rephrased as: “ATP levels in lung homogenates were measured using a commercial assay (Beyotime, China) following the manufacturer’s instructions.”

 Lines 206 Western Blotting: While antibody details are comprehensive, the section lacks information about the imaging system or software used for quantification.

 Line 234 RT-PCR: Mention the specific thermal cycling conditions used for amplification to enhance reproducibility.

Results:

 Well presented.

Discussion:

 The discussion thoughtfully contextualizes the findings within the broader literature.

 It integrates findings with existing literature, particularly the role of oxidative stress, mitochondrial dysfunction, and the antioxidant potential of SS-31. However, the linkage between the experimental results and prior studies can be elaborated further to highlight how this work advances knowledge in the field.

 Incorporate more detailed comparisons between the experimental results and established models of pulmonary fibrosis to contextualize the significance of the observed improvements with SS-31.

 While weight changes, histological evidence, and biochemical markers are valuable, a stronger emphasis on the potential for SS-31 to improve lung function (even if speculative) would be impactful.

 Although the MAPK pathway is mentioned, additional exploration or hypothesizing about other mechanisms (e.g., ROS detoxification efficiency or mitochondrial biogenesis) would add depth to the discussion.

 Consider simplifying some sections to improve readability, especially for readers less familiar with technical details. For instance, the paragraph on mitochondrial dysfunction could benefit from a summary sentence that ties findings to their clinical relevance.

Conclusion:

 This is well written however future studies could benefit from a deeper dive into signaling pathways (e.g., MAPK) to uncover how SS-31 exerts its effects at the molecular level.

References:

 Well documented

Concluding Remarks:

The paper necessitates the aforementioned corrections, inclusive of rectifying any errors highlighted, coupled with a meticulous proofreading by a proficient language expert to refine its English language usage. Upon addressing these amendments, the paper will meet the required standards for acceptance.

Reviewer #2: The manuscript is well written. The purpose, methods and results are clearly presented.Studies involved with the live animals were carried out following protocols approved by the Research Ethics Committee. The statistical analysis in this paper is well conducted.The authors should report some of the limitations of this study in detailed.

6. PLOS authors have the option to publish the peer review history of their article (what does this mean? ). If published, this will include your full peer review and any attached files.

**Do you want your identity to be public for this peer review?** For information about this choice, including consent withdrawal, please see our Privacy Policy .

Reviewer #1: **Yes: ** Dr. Aamir Mushtaq

Reviewer #2: No

---

## [Author Response · Author response to Decision Letter 0]

12 Feb 2025

Dear editor and reviewers

Thanks for revising this study. Our research team fully discussed and revised these comments again, and responded point-by-point to the review comments.

Here is the point-to-point response

Response:We have revised our manuscript format according to your request.

Response: We have added this to the manuscript in part 2.2.

“This research was supported by Heilongjiang Province Key R&D Program(GA21C011) and the National Natural Scientific Foundation of China [82172164] and Heilongjiang Province Key R&D Program(JD22C005).”

Response: We have modified this section according to your opinion.

Response: We have uploaded the experimental data at your request, named “minimal data set”

Response: We have uploaded the experimental data at your request.

6. PLOS requires an ORCID iD for the corresponding author in Editorial Manager on papers submitted after December 6th, 2016. Please ensure that you have an ORCID iD and that it is validated in Editorial Manager. To do this, go to ‘Update my Information’ (in the upper left-hand corner of the main menu), and click on the Fetch/Validate link next to the ORCID field. This will take you to the ORCID site and allow you to create a new iD or authenticate a pre-existing iD in Editorial Manager.

Response: I have added my ORCID iD to my Information at your request.

7. Thank you for stating the following in the Acknowledgments Section of your manuscript:

“This research was supported by Heilongjiang Province Key R&D Program(GA21C011) and the National Natural Scientific Foundation of China [82172164] and Heilongjiang Province Key R&D Program(JD22C005).”

“This research was supported by Heilongjiang Province Key R&D Program(GA21C011) and the National Natural Scientific Foundation of China [82172164] and Heilongjiang Province Key R&D Program(JD22C005).”

Response: We have modified this section according to your opinion.

8. PLOS ONE now requires that authors provide the original uncropped and unadjusted images underlying all blot or gel results reported in a submission’s figures or Supporting Information files. This policy and the journal’s other requirements for blot/gel reporting and figure preparation are described in detail at https://journals.plos.org/plosone/s/figures#loc-blot-and-gel-reporting-requirements and https://journals.plos.org/plosone/s/figures#loc-preparing-figures-from-image-files. When you submit your revised manuscript, please ensure that your figures adhere fully to these guidelines and provide the original underlying images for all blot or gel data reported in your submission. See the following link for instructions on providing the original image data: https://journals.plos.org/plosone/s/figures#loc-original-images-for-blots-and-gels.

Response: We have uploaded the original uncropped and unadjusted images underlying all blot results as a Supporting Information files named “WB”.

Response: We have reexamined this section at your request.

Additional Editor Comments:

Dear Authors

Although this study is intriguing, a few issues need to be fixed before it can be accepted. Please include the study's goal at the conclusion of the introduction. Kindly add the euthanasia technique to the list of procedures.

Response: We have modified this section according to your opinion.

Reviewer #1: Review Report:

The manuscript titled "SS-31 protects against bleomycin-induced lung injury and fibrosis" suggests that the research focuses on the therapeutic potential of SS-31 in preventing or alleviating lung damage and scarring caused by bleomycin in experimental models.

Note: I reviewed the file named (PONE-D-24-54571) downloaded from journal online portal and all my comments are according to the line numberings of the pdf file “PONE-D-24-542471”

Detailed Comments/Recommendations:

Title:

 The title of the study, "SS-31 protects against bleomycin-induced lung injury and fibrosis" effectively communicates the focus of the study, specifically the therapeutic role of SS-31 in addressing bleomycin-induced lung injury and fibrosis. appears to be apt and engaging. It reflects the experimental context and potential outcomes, making it relevant to researchers interested in mitochondrial-targeted therapies or pulmonary fibrosis models.

 The title is suitable for the scope of the study, but slight refinements, such as specifying the experimental model or incorporating mechanistic details, would enhance its clarity, relevance, and appeal to the target audience.

 Suggested Title: SS-31: A Promising Therapeutic Agent Against Bleomycin-Induced Pulmonary Fibrosis in Mice"

Response: We have revised the study title according to your recommendation

Abstract:

 The abstract clearly states the purpose of the research, focusing on the potential protective effects of SS-31 against bleomycin-induced pulmonary fibrosis in mice.

 Lines 57-59: SS-31 was administered daily but does not specify the dosage or route of administration. Including this detail would strengthen the methodological clarity.

Response: We have added the SS-31 administration method and measurement according to your comments

 Lines 62-63: The study demonstrated that SS-31 could potentially mitigate the reduction in mice" is incomplete, leaving the reader unclear about what specific reduction is being referred to.

Response: We have revised the manuscript: BLM can cause a significant decrease in body weight in mice. However, intraperitoneal injection of SS-31 slowed down the body weight loss in mice.

 Lines 62-70: While methods such as HE staining, Masson staining, and Western blot are mentioned, the abstract lacks numerical or statistical details to quantify the findings. Providing key metrics (e.g., percentage reduction in fibrosis or ROS levels) would enhance credibility.

Response: The specific results have been shown in the results, for simplicity, we do not specify in the abstract.

 Lines 71-75: While the abstract mentions outcomes like reduced ROS and inflammatory factors, it does not delve into the specific mechanisms through which SS-31 exerts these effects. A brief mention of the proposed pathways would be helpful.

 Address any typographical errors and ensure proper scientific notation.

Response: Regarding the mechanism of action of SS-31, we have shown in the discussion that, for simplicity, we did not specify it in the abstract.

We have re-checked the manuscript at your request to standardize the scientific notation.

Key Words: Three out of five key words are already mentioned in title. It will be more attractive if the writer mentions some other relevant key words.

Response: We have added the keyword "mitochondria" according to your suggestion.

Introduction:

 Combine the 1st and 2nd paragraphs to provide a concise yet comprehensive summary of pulmonary fibrosis and its challenges.

 In 2nd Paragraph: The section on pulmonary fibrosis repeats some information (e.g., limited treatments and poor prognosis). Streamlining these details can make the introduction more concise.

Response:We have modified this section according to your suggestion.

 In 2nd paragraph: While the role of oxidative stress and ROS in pulmonary fibrosis is mentioned, the discussion could benefit from a brief elaboration on how SS-31 specifically counteracts these processes.

Response: We have modified this section according to your suggestion.

 Lines 107 to onward: Include a specific mention of how SS-31 reduces ROS or preserves mitochondrial integrity, tying it directly to pulmonary fibrosis pathogenesis.

Response: We have modified this section according to your suggestion.

 Lines 122 to onward: Clarify the connection between pulmonary hypertension and fibrosis, or reduce the emphasis on PH if it is not a major focus of the research.

Response: We have removed this part based on your recommendation.

 Some references (e.g., oxidative stress role, SS-31’s effects) are summarized without detailed explanations or direct linkage to the study’s aims. Consider elaborating on key studies to provide stronger support for SS-31's potential.

Response: We have removed this part based on your recommendation.

Materials and Methods:

 Lines 139-141: “The killing methods and model establishment methods have passed the ethical standards...”: Rephrase to improve readability and remove redundancy.

Response:We have modified this section according to your suggestion.

 Lines 151-153: Terms like “molding” (e.g., “from 1 day before molding until day 28 after molding”) are unusual and may confuse readers. Use “model induction” or “bleomycin treatment” instead.

Response:We have modified this section according to your suggestion.

 Lines 162, Histopathology: The paraffin embedding and sectioning steps should clarify equipment or specific dehydration/embedding protocols.

Response: We have modified this section according to your suggestion.

 Lines 192-193: ATP content in mouse lung homogenate was determined according to commercial protocols” can be rephrased as: “ATP levels in lung homogenates were measured using a commercial assay (Beyotime, China) following the manufacturer’s instructions.”

Response: We have modified this section according to your suggestion.

 Lines 206 Western Blotting: While antibody details are comprehensive, the section lacks information about the imaging system or software used for quantification.

Response: We have modified this section according to your suggestion.

 Line 234 RT-PCR: Mention the specific thermal cycling conditions used for amplification to enhance reproducibility.

Response: We have modified this section according to your suggestion.

Results:

 Well presented.

Discussion:

 The discussion thoughtfully contextualizes the findings within the broader literature.

 It integrates findings with existing literature, particularly the role of oxidative stress, mitochondrial dysfunction, and the antioxidant potential of SS-31. However, the linkage between the experimental results and prior studies can be elaborated further to highlight how this work advances knowledge in the field.

Response: We have modified this section according to your suggestion.

 Incorporate more detailed comparisons between the experimental results and established models of pulmonary fibrosis to contextualize the significance of the observed improvements with SS-31.

Response: We have modified this section according to your suggestion.

 While weight changes, histological evidence, and biochemical markers are valuable, a stronger emphasis on the potential for SS-31 to improve lung function (even if speculative) would be impactful.

Response: Thanks for your suggestion, this suggestion is very constructive.We also included this part in the limitations of our discussion section. However, we could not directly detect the lung function of C57BL /6 mice. It can only be proved indirectly by histology and by biological indicators. In future experiments, we will consider using rats or larger experimental animals, at which time we can directly test the effect of SS-31 on lung function.

 Although the MAPK pathway is mentioned, additional exploration

---

## [Decision Letter · Decision Letter 1]

3 Mar 2025

SS-31: A Promising Therapeutic Agent Against Bleomycin-Induced Pulmonary Fibrosis in Mice

PONE-D-24-54571R1

Dear Dr. Zhao,

We’re pleased to inform you that your manuscript has been judged scientifically suitable for publication and will be formally accepted for publication once it meets all outstanding technical requirements.

Kind regards,

Sairah Hafeez Kamran, PhD

Academic Editor

PLOS ONE

---

## [Editor Report · Acceptance letter]

PONE-D-24-54571R1

PLOS ONE

Dear Dr. Zhao,

I'm pleased to inform you that your manuscript has been deemed suitable for publication in PLOS ONE. Congratulations! Your manuscript is now being handed over to our production team.

Kind regards,

on behalf of

Dr. Sairah Hafeez Kamran

Academic Editor

PLOS ONE